

# Cracking hadron and nuclear collisions open with ropes and string shoving in PYTHIA8

**Smita Chakraborty**⋆

Lund University, Sölvegatan 14 A, Lund 223 62, Sweden

⋆ smita.chakraborty@thep.lu.se

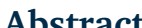

*50th International Symposium on Multiparticle Dynamics (ISMD2021)
12-16 July 2021*

## Abstract

Enhancement in strangeness yields in heavy-ion collisions is regarded as one of the attributes of the Quark-Gluon Plasma (QGP) along with jet quenching and final-state collectivity. To reproduce these effects in the MC implementation of the Lund string model in PYTHIA, the inclusion of non-perturbative mechanisms like string shoving and rope hadronization is crucial. In this proceeding, we discuss our latest implementation of these mechanisms and the preliminary results for the strangeness enhancement in high-multiplicity p-p, p-Pb and Xe-Xe. We conclude with remarks on the ongoing development of the model.

## 1 Introduction

Efforts to understand QGP signatures using QCD models have been an active field since the first observation of near-side long-range two-particle correlations in high-multiplicity p-p collisions at the LHC in 2010. On the other hand, tools like Monte Carlo event generators, particularly PYTHIA, have been quite successful at describing LEP physics and p-p phenomenology. Such nuclear effects like final-effect collectivity in p-p collisions sparked the need to probe both p-p and nuclear collisions more closely using string dynamics. Currently, nuclear collisions can be generated within PYTHIA using the Angantyr framework [1] which uses the p-p MPI framework in PYTHIA along with an advanced version of the Glauber model. In recent times, non-perturbative processes such as string interactions, called string shoving [2,3] and rope hadronization [4], have been studied to generate QGP-like signals in all collision systems within PYTHIA8. Such analyses come coupled with the challenges of formulating an approach that can treat both high and low multiplicity events with the same physical picture and also treating all partons in an event equally, regardless of their transverse momentum $p_\perp$.

To include string interactions in a whole event, a new approach with a Lorentz frame, called the parallel frame, has been constructed that accommodates all strings formed in an event, regardless of the $p_\perp$ of the partons forming the strings. In a PYTHIA event, every possible pair of string pieces are boosted to this parallel frame, where we calculate the repulsive force a.k.a. shoving between the strings and hadronize the strings via rope formation. They are then boosted back to the laboratory frame where the successive processes, such as hadron decay and rescattering, follow. This is a novel tool that has the capacity to reproduce the QGP signals such as strangeness enhancement and collective flow in both small and large systems.

## 2 Review of previous implementations

The first implementation of rope hadronization in the Lund class of event generators was done in DIPSY, where the subsequent hadronization was done by PYTHIA8 [5]. The primary idea in rope formation is that when two or more strings are close enough in the transverse coordinate space, the colour charges of the partons at the string ends combine to form higher colour multiplets with a wider colour fluxtube called rope [4]. As a result of this rope formation, the string tension $\kappa$ increases between the endpoint charges, which we call $\kappa_{\text{eff}}$. When this rope hadronizes, each string in the rope hadronizes *separately* but not independent of each other because of the modified string tension $\kappa_{\text{eff}}$. When a breakup occurs in such a string, the higher colour multiplets are reduced to lower multiplets *releasing* energy. This energy is then available for the production of strange quarks via the tunneling mechanism which follows the production probability $\mathcal{P}$:

$$\frac{d\mathcal{P}}{d^2 p_\perp} \propto \kappa_{\text{eff}} \exp\left(-\frac{\pi}{\kappa_{\text{eff}}}(\mu^2 + p_\perp^2)\right), \tag{1}$$

where $p_\perp$ is the transverse momenta of the $q\bar{q}$ produced and $\mu$ is the mass of the produced quark. Borrowing from lattice calculations [7], that the string tension is proportional to the quadratic Casimir operator $C_2$ [5], the modification of the string tension is given by the equation:

$$\kappa_{\text{eff}} = \kappa^{\{p+1,q\}} - \kappa^{\{p,q\}} = \frac{2p + q + 4}{4}\kappa, \tag{2}$$

where the colour field in a rope transitions from $\{p+1,q\}$ state to $\{p,q\}$ and $\kappa$ is the string tension in a single string.

The other non-perturbative string interaction that we consider here is the string shoving. String shoving is the repulsive force generated between two strings when they are close to each other in the three-dimensional coordinate space. To calculate the shoving force, the colour electric field in a string of radius $R$ is considered to have a Gaussian form:

$$E(r_\perp) = C \exp\left(-\frac{r_\perp^2}{2R^2}\right), \tag{3}$$

where $r_\perp$ is the perpendicular distance from the center of the string and $C$ is a constant. For two such cylindrical strings lying parallel to each other, the resulting interaction force is given by:

$$f(d_\perp) = \frac{dE_{\text{int}}}{dd_\perp} = \frac{g\kappa d_\perp}{R^2} exp\left(-\frac{d_\perp^2(t)}{4R^2}\right), \tag{4}$$

where $d_\perp$ is the transverse separation between the two strings, $\kappa$ is the string tension and $g$ is a tunable parameter ($\sim \mathcal{O}(1)$). Finally, this shoving force would add over time intervals for each hadron and the final $p_\perp$ is distributed to them after hadronization. The first implementation [2] is an oversimplified case where only the strings parallel to each other and to the beam axis were considered for computing the shoving force.

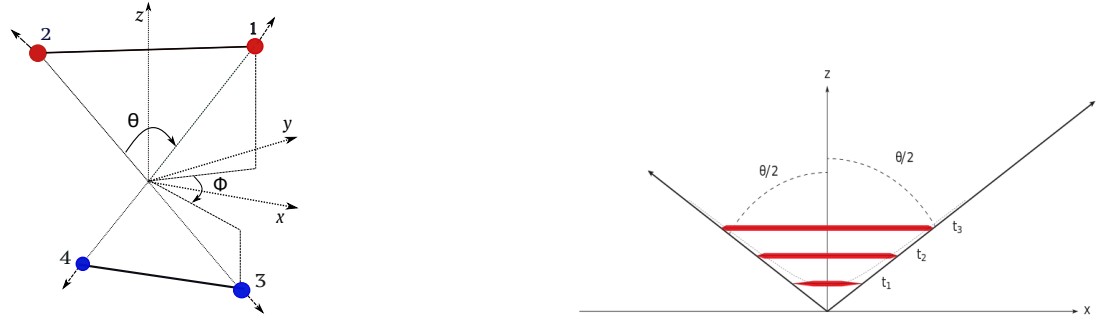

Figure 1: Left: Parallel frame with opening angle $\theta$ and skewness angle $\phi$, right: Evolution of a string in the parallel frame

## 3 New implementation of string shoving and rope hadronization

### 3.1 The *parallel* frame

For a more realistic approach with high $p_\perp$ partons in dense environments where the majority of all strings produced are neither parallel to the beam axis nor to each other, we boost to a Lorentz frame where the strings lie in parallel planes. The novelty of this work is the implementation of string shoving and rope hadronization using the parallel frame formalism. The usefulness of the parallel frame are two folds:

- In the parallel frame, every pair of string pieces between which the repulsive force is calculated, lie in parallel planes with respect to each other, hence maximizing the symmetry in their topology which makes the calculation of shoving force easier,

- Since we boost every possible string pair from the lab frame to the parallel frame, even high $p_\perp$ partons, like jets, are considered in the calculation of the shoving force. This considers the shoving force between all possible string pairs which is crucial in heavy-ion collisions where a large fraction of the strings formed are neither parallel to each other nor parallel to the beam axis.

In the parallel frame, as shown in figure 1, each string evolves in width until it reaches the maximum width $R$ ($\sim 1$ fm). The pseudorapidity $\eta$ and skewness angle $\phi$ between the string pair in the parallel frame can be expressed in the form:

$$\cosh\eta = \frac{s_{13}}{4p_{\perp 1}p_{\perp 3}} + \frac{s_{14}}{4p_{\perp 1}p_{\perp 4}} \ \text{and} \ \cos\phi = \frac{s_{14}}{4p_{\perp 1}p_{\perp 4}} - \frac{s_{13}}{4p_{\perp 1}p_{\perp 3}}, \tag{5}$$

where $s_{ij}$ are the squared masses for the partons $i$ and $j$, $p_{\perp i}$ is the transverse momentum of the parton $i$ in the parallel frame with respect to the x-axis [6]. For further details on the shoving implementation, we refer the reader to [3].

In the new rope implementation, once in the parallel frame, we calculate the repulsive force between two string pieces while they evolve to reach the maxmimum width $R$. Once they have reached $R$, the spatial overlap between the string pair is calculated. If there is a non-zero overlap, the string ends merge to form higher colour multiplets and the strings coalesce to form *ropes* in the parallel frame. This mechanism is motivated by the spread in the colour flux tubes when two strings lie close to each other in the transverse coordinate space [7]. Rope

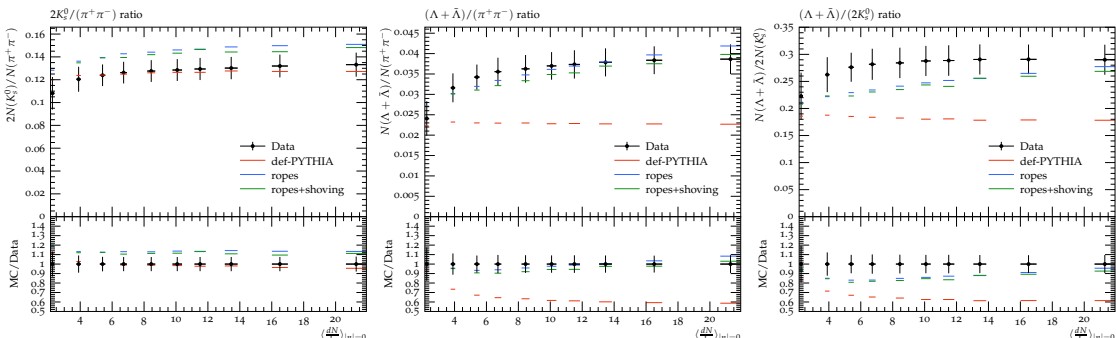

Figure 2: Yield ratios of strange hadrons compared to ALICE data for p-p events at $\sqrt{s}$=7 TeV: $2K_s^0/(\pi^++\pi^-)$ (left), $(\Lambda+\bar{\Lambda})/(\pi^++\pi^-)$(center) and $(\Lambda+\bar{\Lambda})/2K_s^0$(right) vs. $\langle dN/d\eta \rangle$.

formation poses sufficient effect on the string tension $\kappa$ in dense environments like in high multiplicity p-p, p-A and A-A collisions, to have a modified string tension $\kappa_{\text{eff}} > \kappa$. During hadronization, when such higher multiplet transitions to a lower colour multiplet, energy is *released* which is then accounted for via the modified fragmentation parameters influenced by $\kappa_{\text{eff}}$ for each string break-up. Hence the strings hadronize *separately* with modified string fragmentation parameters, thereby not hadronizing independently anymore.

## 4 Preliminary results

In this section, we present some preliminary results based on the implementation described in section 3. We first present the strange hadrons to pion yield ratios in p-p collisions and compare that with ALICE data. In the next subsection, we present the strangeness yields obtained from a RIVET analysis without comparison to data, where we show the the yield ratios of strange hadrons to pion in p-p, p-Pb and Xe-Xe.

### 4.1 Comparison to ALICE data

We have generated 10 million minimum bias p-p events at $\sqrt{s} = 7$ TeV with Monash tune values [8] and compared to ALICE data at mid-rapidity $|y| < 0.5$ [9]. We look at the production yield ratios of $K_s^0$ and $\Lambda$ to $\pi^++\pi^-$ over $\langle dN/d\eta \rangle$ in figure 2. In the ratio of $2K_s^0$ to $\pi^++\pi^-$ (left), we notice that ropes with and without shoving, overshoot the data whereas default PYTHIA provides a good agreement to data. This is because rope hadronization and shoving mechanisms generate fewer pions in general. The ratio of $\Lambda + \bar{\Lambda}$ to $\pi^++\pi^-$ (center) shows more clearly that the baryon yield in the new implementation is higher than default PYTHIA, which is reinforced by the plot on the right in figure 2, where ratio of $\Lambda + \bar{\Lambda}$ yield to $K_s^0$ is shown.

### 4.2 Strangeness yields in p-p, p-Pb, and Xe-Xe collisions

Using a RIVET analysis, we study ALICE primary particles and count the yield of $\Lambda$ and $\bar{\Lambda}$ particles and plot the ratio to the total yield of $\pi^+$ and $\pi^-$ at mid-rapidity $|\eta| < 0.5$. In figure 3, we have the yield ratios for p-p collisions at 7 TeV, p-Pb collisions at 5.02 TeV and Xe-Xe collisions at 5.44 TeV plotted against charged particle multiplicity($N_{\text{ch.}}$) in $|\eta| < 0.5$. We note that the yield ratios with the new rope implementation, with and without shoving, increases monotonously while going from p-p to p-Pb and Xe-Xe. Work to compare these yields

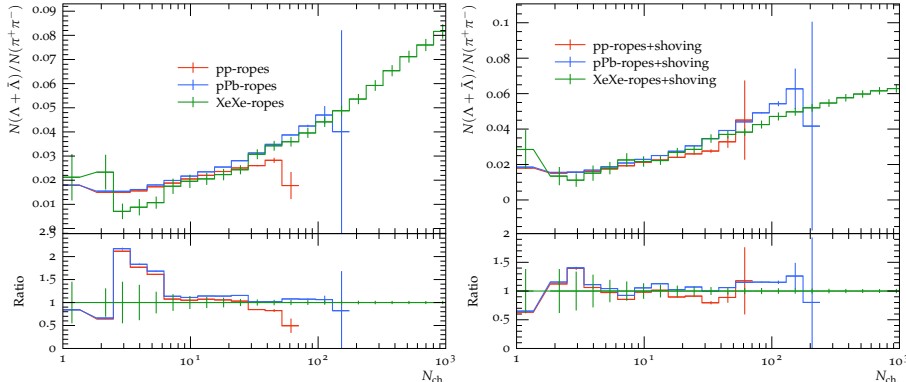

Figure 3: Plots for $(\Lambda + \bar{\Lambda})/(\pi^+ + \pi^-)$ for p-p at 7 TeV, p-Pb at 5.02 TeV and Xe-Xe at 5.44 TeV with the new rope implementation, without shoving (left) and with shoving $g = 0.25$ (right).

to experimental data are currently in progress.

In the figure 4, we compare the $(\Lambda + \bar{\Lambda})/(\pi^+ + \pi^-)$ yields for p-Pb (left) and Xe-Xe (right) Angantyr, with ropes only and ropes with shoving $g = 1$ plotted against $N_{ch.}$ in $|\eta| < 0.5$. We note that the yield ratios with the new rope implementation, both with and without shoving, are 20% more compared to default Angantyr at lowest multiplicities and increases with $N_{ch.}$.

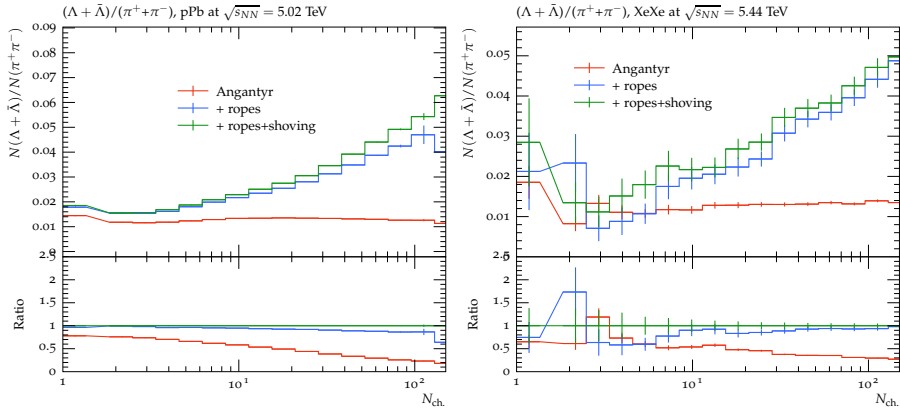

Figure 4: Yields of $(\Lambda + \bar{\Lambda})/(\pi^+ + \pi^-)$ for p-Pb at 5.02 TeV (left) and Xe-Xe at 5.44 TeV (right).

**Note on tuning of fragmentation parameters:** We would need to tune fragmentation parameters since the ratio of $s/u$ quarks in default PYTHIA are tuned to LEP results [10,11]. Also, the magnetic moment contribution from strange quarks affects the fragmentation parameters which has not been accounted for in the previous rope implementation [5]. Hence, to maintain the good description of the data with the current rope implementation including the strangeness magnetic moment, tuning to the average $s/u$ ratio for several experiments where this value has been calculated to a higher precision e.g. in [12,13], is required.

# 5 Conclusion

To conclude, we see that implementing both string shoving and rope hadronization in PYTHIA8/Angantyr with the parallel frame formalism can give rise to strangeness enhancement. The novelties in this approach are many - generation of QGP signals in high-multiplicity p-p, p-A and A-A collisions and inclusion of jets in all systems - even though we do not explore the latter case in this proceeding. Studies to check if this approach can produce experimental observations and predict signals like jet quenching in small systems are still underway. Hence, to simulate a heavy-ion event in PYTHIA/Angantyr, inclusion of such Lund string interactions are necessary.

# Acknowledgements

This work is done in collaboration with Christian Bierlich, Gösta Gustafson and Leif Lönnblad. This work has received funding from the European Union's Horizon 2020 research and innovation programme as part of the Marie Skłodowska-Curie Innovative Training Network MC-netITN3 (grant agreement no. 722104).

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
