# Peer review of "Cracking hadron and nuclear collisions open with ropes and string shoving in PYTHIA8"

_SciPost Physics Proceedings, doi:SciPost Phys. Proc. 10, 017 (2022)_

## Round 1 · Referee Report · Anonymous · 2022-1-28

Strengths

An excellent explanation of the background, and of the new developments.

Weaknesses

Nothing to exclude from proceedings publication.

The role of the novel parallel-frame development (which displays as the ropes -> ropes+shoving?) isn't made so clear, with the main effect being the addition of ropes to Pythia default.

There is some lack of clarity about the extent to which strangeness production is enhanced, vs. baryon production. Maybe this can be disambiguated more in future work.

Figs 3 and 4 don't seem to clearly define the x-axis... I am not sure what it is, and hence what the shoving is scaling with! Some event energy-scale variable, e.g. sumET?

Report

Very good proceedings write-up. I am happy to publish as is, but if the authors want to update with the Fig 3 and 4 x-axis clarified (or point at the explanation I missed) that would be a useful improvement.

---

## Editorial Decision

published